# Relation among Caregivers’ Burden, Abuse and Behavioural Disorder in People with Dementia

**DOI:** 10.3390/ijerph18031263

**Published:** 2021-01-31

**Authors:** Ignacio Gimeno, Sonia Val, María Jesús Cardoso Moreno

**Affiliations:** 1Department of Psychology and Sociology, Faculty of Health of Sciences, University of Zaragoza, 50009 Zaragoza, Spain; nachetegimeno@hotmail.com; 2EINA, Design and Manufacturing Engineering Department, University of Zaragoza, 50009 Zaragoza, Spain; sonia@unizar.es

**Keywords:** dementia, informal caregivers, burden, caregiver’s abuse, behavioural disorders

## Abstract

Dementia produces a loss of independence to carry out the activities of daily life. The great demand for care that these people need usually falls on the family through informal care. This study aims to analyse the burden showed by the informal caregiver of a person with dementia. In addition, we analyse whether this burden present in informal caregivers could be related to abusive behaviour. We also study the relationship between the stage of the disease, the appearance of behavioural disorders and the level of burden in the caregiver using the Scales of Zarit, CASE and FAST. The data showed that 45.50 per cent of caregivers have light burden or burden. After the research, it was identified that the presence of behavioural disorders in patients with dementia showed a correlation with the increase in both the main caregiver burden and abuse. An increase in the level of burden is followed by an increase in the level of abuse (r = 0.844; *p* = 0.000). Furthermore, we analysed several conditions that could have a correlation with this burden and abuse. It was found that burden in the caregiver could be linked with the presence of behavioural disorders, like aggression (r = 0.577; *p* = 0.008) and irritability (r = 0.600; *p* = 0.005) at the moderate stage of the disease. On the other hand, there is a positive correlation between the probability that people with dementia suffer abuse in the moderate stage of the disease and the presence of aggression (r = 0.732; *p* = 0.000), lack of inhibition (r = 0.571; *p* = 0.009) and irritability (r = 0.827; *p* = 0.000). Taking this data into account, burden and abuse seem to be linked to the presence of behavioural disorders in patients with dementia in the moderate stage.

## 1. Introduction

The scientific and technological advances generated in recent decades, especially in developed countries have allowed the extension of life and have helped older people to fully enjoy the final stages of their lives. The increase in life expectancy and aging of the population inevitably favours the appearance of those diseases in which age is a fundamental risk factor, as is the case with dementia. The number of people with dementia worldwide was estimated in 2016 at 46.8 million. However, the increase in the elderly population, especially of people over 80 years old, in which the risk of dementia is accentuated, added to the absence of an effective treatment and the difficulty of an early diagnosis, can anticipate that this figure will be doubled every 20 years, so it is estimated that will affect 131.5 million people in 2050. Therefore, dementia is a real public health problem worldwide, with a high economic and social cost, being considered one of the epidemics of the 21st century [1,2]. 

Dementia is, moreover, one of the conditions that most limits our ability to conduct daily activities, significantly reducing the quality of life and autonomy of the patient. Associated with a progressive, global and irreversible cognitive deterioration, dementia causes a loss of memory and other higher cognitive functions. It also leads to alterations in behaviour and personality with important repercussions on the patient functional capacity [3,4,5].

This disease often results in a situation of complete physical and psychological dependence and create a need for continuous supervision and care as it evolves. However, the imbalance between the health demands generated by the growing increase in cases of dementia and the insufficient resources available, have meant that the care and attention to these dependent elderly people has often fallen to the family, on which the impact is intensely negative [6,7,8].

The ties of affection and emotional closeness with the sick person mean that, despite the absence of remuneration and training for care, it is carried out with great commitment and dedication by the family. However, not all members of the patient’s family or surroundings take on the responsibility of carrying out this arduous task in the same way [9,10]. 

The World Health Organization indicates certain characteristics of the informal primary caregiver: usually a person close to the patient, often a female, who cares for the patient regularly at home, in difficult conditions and without remuneration. The caregiver generally covers the basic needs of the patient, on many occasions showing a lack of training and skills to perform their function [11]. 

The main caregiver, therefore, supports the mental, physical and socioeconomic weight in the management of the sick person throughout the development of the disease. Their task requires time, energy and effort, significantly influencing their quality of life. In addition, they assume new responsibilities for which, generally, they have not received any training and which are, sometimes imposed by the family, often as a consequence of the lack of economic resources. All this, added to the length of time the disease continues for, contributes to the creation of a chronic stress situation in the caregiver known as “burden” [12,13,14]. 

Burden, therefore, is the result of physical-emotional work and social constraints generated by the tasks necessary for the care of a sick person. It has negative repercussions on the physical (cardiovascular and immune) and mental (anxiety, depression, anger, exhaustion, fatigue, tiredness) health of the caregiver [15]. How well the caregiver copes the situation, their motivation for care, the support networks available, and the caregiver-affective relationship are decisive in the appearance of burden [16,17,18]. 

On the other hand, the loss of autonomy, the degree of cognitive deterioration of the patient with dementia, their physical manifestations and, especially, their behavioural manifestations, also influence burden [19,20,21]. However, some authors such as Van der Lee et al. [22] place manifestations or behavioural disorders as the aspect of dementia that has the greatest impact on caregiver burden.

The effect of these behaviours is of such magnitude, that they constitute 50 per cent of the problems that arise in dementia consultations, being one of the aspects of the disease that most worries the family and that most complicates the care of these people and sometimes becoming a reason for marginalization, institutionalization and even ill-treatment by the caregiver. The presence of behavioural disorders generates high levels of stress in the caregiver and is associated with the appearance of physical and psychological illnesses in 50 per cent of them, negatively affecting both the quality of life of the caregiver and the quality of life of the person with dementia by reducing the quality of the care they give [23,24,25]. 

One of the consequences that the task of caring for a person with these characteristics can have is the presence of abuse exercised by the main caregiver towards the person with dementia. Some studies show an abuse rate of up to 50 per cent in these situations. In this context, the abuse of the elderly is defined in The United Nations Second World Assembly on Ageing [26] as “any single or repeated act, or lack of appropriate action that occurs in any relationship supposedly of trust, that causes damage or distress to an elderly person”. 

Abuse is considered, therefore, as a type of violence in which, from the point of view of the ecological model, it is the result of the interaction of (1) individual factors (impulsivity, having suffered abuse, low educational level etc.), (2) relational factors (the way of relating to friends, family, or the couple), (3) relating to community factors (school, workplace and neighbourhood), and (4) social factors (cultural norms, or social policies that maintain socioeconomic inequalities). However, the presence of burden in the caregiver and the manifestation of behavioural alterations by the patient with dementia, added to the inability of many of these people to report abuse and lack of social support can also be critical [27,28,29]. 

The progressive and disabling cognitive impairment associated with the development of dementia puts these patients at risk of experience some or several of the different forms of abuse mentioned. In addition, some studies [30,31] link a greater occurrence of abuse to the context of informal care because of the complexity of identifying and intervening in these abusive situations. Firstly, for the difficulty of the patient with dementia to report the abuse. Secondly for the social isolation they face and thirdly the denial of the situation that may exist by the caregiver. 

Another aspect to be considered is that the presence of behavioural disorders in the person with dementia can cause an increase in caregiver´s burden. In turn, we must bear in mind that these behavioural disorders do not manifest themselves uniformly throughout the disease [32]. As a consequence, we can deduce that the greater the intensity of behavioural disorders, the greater the overload on the caregiver. These reasons in addition to the lack of awareness by health professionals, place these non-institutionalized patients in a situation of special vulnerability. Therefore, violence against people with dementia in the context of informal care, represents a real vital risk and a source of suffering and anguish for the patient and his family.

Based on the above, one of the hypotheses of our work is that caregivers of people with dementia suffer a high degree of burden. In addition, it is considered that the greater the degree of burden, the greater the probability of manifesting abusive behaviour. It must also be highlighted the fact that the appearance of behavioural disorders could cause a greater burden on the caregiver and increase the risk of abuse. 

In this regard, the objective of this research is to analyse the burden showed by the informal caregiver of a person with dementia. In addition, we intend to verify whether this burden present in informal caregivers would be related to abusive behaviour. We also study the relationship between the stage of the disease, the appearance of behavioural disorders and the level of burden in the caregiver.

## 2. Materials and Methods

### 2.1. Participants 

The sample of this study was composed of 33 informal caregivers of patients with dementia residing in a Spanish city. The inclusion criterion used for the selection of the participants in this study was that they deal with people diagnosed with dementia in any of its forms whose care was exercised only by the informal caregiver.

People with dementia who were institutionalized and those situations in which the care of the person with dementia was performed by a professional caregiver were excluded from the study. Regarding non trained caregivers, those who, due to the existence of physical, cognitive or cultural barriers could not correctly carry out the questionnaires necessary for the development of this research, were excluded.

### 2.2. Variables and Instruments

#### 2.2.1. Variables

Two types of variables were used in the study: independent and dependent. The dependent variables were demographic data, personal data and the phase of dementia in which the user is. The independent variables were:Burden with three level: non burden, light burden and burden. The instrument chosen for its measurement was Caregiver Burden Interview (ZARIT)Abuse with two levels: non abuse and abuse. The instrument chosen for its measurement was Caregiver Abuse Screen (CASE).Behavioural disorder with 12 different levels according to each behavioural disorder. The instrument chosen for its measurement was Behavioural Disorders Questionnaire.

#### 2.2.2. Instruments

##### Caregiver Burden Interview (ZARIT)

It is the most widespread questionnaire to assess the level of burden experienced by informal caregivers of people with dementia and other disorders. It was developed by Zarit and his collaborators in 1980 [33] and later adapted to the Spanish population by Marín et al. en 1996 [34]. It allows users to obtain a multidimensional view of the mental, social and physical state of the caregiver, including the dimensions of quality of life and self-care capacity, social support network and competencies to face behavioural and clinical problems of the patient care. Although there are different versions, the one employed in this study [34] consists of 22 questions like “Likert” with 5 options, in which the caregiver must indicate how often he or she feels stress, being (0) never, (1) with rarely, (2) sometimes, (3) often and (4) almost always. These results are added up to a total score that can range from 0 to 88 points. The questionnaire has good psychometric properties and allows them to discriminate between different levels of burden, classifying the situation of the caregiver into: “absence of burden” if the score obtained is under 46 points, “light burden” if the score is between 47 and 55 points or “burden” if the score is higher than 55 points. The internal consistency of the scale is 0.91 and realiability tes-retest is 0.96 [35,36].

##### Functional Assessment Staging (FAST)

It is a functional assessment classification scale [37] created by Reisberg in 1988. which has been designed to assess changes in functional performance and the activities of the user’s daily life throughout all stages of dementia. These changes are closely related to the progressive cognitive decline associated with the development process of this disease. The evaluation procedure consists of the description of 16 successive stages ranging from a situation of functional normality to the levels of dependence and functional impairment associated with the most advanced phases of the disease and thus relating the patient with the stage that best reflects the situation of the patients. Functional characteristics described above are associated with clinical diagnosis as normal (stage 1), subjective cognitive impairment (stage 2), mild cognitive impairment (stage 3) mild dementia (stage 4), moderate dementia (from stage 5 to stage 9) and severe dementia (from stage 11 to stage 16). The Consistency of this instrument is 0.86 and the validity is 0.79 [37]. 

##### Caregiver Abuse Screen (CASE)

It is a validated screening tool which serves as the first warning to detect the risk of abuse, psychosocial abuse or neglect by the caregiver towards the patient with dementia. It was created by Reis and Nahmiash in 1995 [38] and is validated in Spain by Rivera-Navarro et al. [39]. It contains 8 questions with dichotomous answers about situations related to violence as cited. A score of 4 or more points on the scale would reflect the existence of abuse. The CASE scale has a moderate internal consistency (α = 0.75) [39]. The reliability of this scale is 0.696 [40].

##### Behavioural Disorders Questionnaire

This is a questionnaire developed for this study that aimed to collect information provided by caregivers about behavioural disorders present in patients with dementia.

This questionnaire shows 12 different behavioural alterations or disorders that can be detected in patients with dementia, along with a brief explanation of some behaviour commonly associated with such alterations written with colloquial words for better understanding. The caregiver should indicate how often the patient exhibits these types of behaviour, with a score ranging from zero to ten (being zero “never” and ten “daily”). The presence of the symptom and its manifestation was defined based on the classification of the Neuropsychiatric Inventory (NPI) [41]. The NPI-Q Spanish version showed strong test-retest reliability for total symptom scale and for distress scale, besides convergent validity with NPI total symptom (r = 0.879). The different behavioural alterations studied are:Aggression, such as any physical or verbal behaviour that may cause physical or moral/mental harm; unjustified opposition or resistance; agitation; cries; rage; violence; constant criticism, etc.Depression, such as constant sadness; loss of interest or satisfaction in almost all activities; feeling like it’s a burden; lack of hope; demotivation; isolation, etc.Anxiety, such as fear or feeling of loss of control, expressed verbally, through gestures or through bodily language; fear; nervousness; panic, etc.Euphoria, such as abnormally high or inappropriate humour for the given time or situation, without apparent cause; he’s suddenly very happy for no apparent reason, etc.Apathy, such as lack of interest; lack of motivation, feeling, emotion or concern; he finds himself passive, he does nothing; everything doesn’t matter to him/her, etc.Lack of inhibition, such as lack of social touch in language, body expression or other behaviour; acts impulsively, etc.Irritability, such as bad mood; unjustified rapid mood swings; impatience; intolerance; responds badly for no reason; everything makes him angry, etc.Motor hyperactivity, such as increased wandering or other activity that is not explained by a cause; attempts to escape; the patient follows the caregiver constantly; walks incessantly, etc.Repeated vocalizations, such as repeated annoying sounds or vocal expressions; repeats or incessantly asks for the same thing; says someone’s name constantly, etc.Decay, such as loss of interest; dissatisfaction in almost all activities; feeling like it’s a burden; lack of motivation; lack of relationship with others; constant sadness, etc.Passivity, such as the patient do not want to perform any activity; everything doesn’t matter to him/her; lack of motivation; indifference, etc.Anguish, such as feeling of loss of control, expressed verbally, gesturally or through body movements, etc.

In order to give greater validity to this questionnaire, descriptions of similar behaviours were used among some disorders of which were depression and decay, apathy and passivity, and finally anxiety and distress. Obtaining similar values among these behavioural alterations reflects a correct execution and understanding of the questionnaire by the caregiver.

##### Demographic Data

The Personal Questionnaire collects personal and demographic data related to the primary caregiver and the patient, such as gender, caregiver-patient kinship, age, time of disease progression, type of dementia, and time the caregiver has been acting as the primary caregiver.

### 2.3. Procedure

This study obtained favourable opinion for its implementation by the Research Ethics Committee of Aragón (CEICA) with Ethical approval code PI18/402, 30 January 2019. The search for the participants was made by a provisional list provided by the Computer Service of the Lozano Blesa University Clinical Hospital of Zaragoza, with the relevant permission of the Medical Coordination. We contacted by telephone and/or face to face with 67 caregivers. 

The inclusion criteria used for the selection of the participants in this study are: they were users diagnosed with dementia in any of its forms, and their care was exercised exclusively by informal caregivers. 

Following were considered exclusion criteria: the user was institutionalized and care was carried out by formal caregivers. The existence of physical, cognitive or cultural barriers that prevent the caregiver from correctly completing the questionnaires was also considered as an exclusion criterion.

4 out of 37 caregivers who met the inclusion/exclusion criteria decided to not participate in the study. Finally, the sample of our study was composed of 33 caregivers that signed the permission to participate in the study. The Zarit Scale, the Caregiver Abuse Scale, and the Behavioural Disorders Questionnaire were conducted by the primary caregiver at home or with the help of the interviewer, depending on the abilities and preferences of the caregiver. After that, we collected all the data. FAST data were obtained from the patient´s medical history.

### 2.4. Data Analysis

All analyses were carried out using the statistical package IBM SPSS 23 (IBM Corp: Armonk, NY, USA). Arithmetic mean, standard deviation and percentages were analysed. Then, we used Kolmogorov-Smirnov test whether the dependent variables conformed to a normal distribution. As the variables turned out to conform to a normal distribution, different parametric tests were used (Pearson correlation test, t-Student and ANOVAs).

Among the demographic and personal data, the sex and age of both the caregiver and the person with dementia, the kinship of the caregiver with the patient, the type of dementia in the patient and the stage of dementia in which they are found were recorded. Also, we analysed patient’s level of functional impairment, caregiver burden, abuse and behavioural disorders. Continuous type variables are represented as the mean and standard deviation, while categorical variables are represented as percentages and absolute frequencies. Pearson’s correlation coefficient has been used to analyse the linear dependence between burden, abuse, behavioural disorder and stage of disease variables. Differences between the mentioned variables were investigated using Student´s t-test. Statistical significance was set at *p* < 0.05(two-tailed). 

## 3. Results

### 3.1. Demographic Data

The sample was composed by 33 informal caregivers of patients with dementia. Table 1 represents the percentages relating to sex and age, both of the caregiver and the person with dementia, the kinship of the caregiver with the patient, the type of dementia that the stage of dementia.

### 3.2. Caregiver Burden Analysis

Following the analysis of the data obtained in the Zarit questionnaire, it is observed that 45.50 per cent of caregivers have light burden or burden (Table 2).

When we analyzed caregiver burden based on gender, we found that women presented higher levels of burden than men (t = 2.31; *p* = 0.027) (Table 3).

### 3.3. Caregiver Abuse Screen (CASE)

When data obtained from the Caregiver Abuse Screen are analyzed, we found that in more than half of the cases, caregivers presented behaviours related to abuse. (Table 4). No significant differences in abuse were found according to sex (t = 0.790; *p* = 0.430). 

### 3.4. Behavioural Disorders

The following table shows the intensity with which behavioural disorders appeared in people with dementia (Table 5).

As it can be seen, apathy and depression are the disorders that appear most frequently.

### 3.5. Burden and Abuse

Another of the objectives set out in this study was to find out if the caregiver’s burden might be related to abuse. The Pearson correlation test were done between the presence of burden in the caregiver and abuse towards the people with dementia. The results obtained reflect the existence of a strong positive correlation between these variables (r = 0.844; *p* = 0.000) indicating that a higher level of burden correlated with abuse by caregiver (Table 6).

### 3.6. Burden and Behavioural Disorders

A significant positive relationship has been obtained between burden and the presence of behavioural alterations of aggression, depression, anxiety, euphoria, apathy, lack of inhibition, irritability, motor hyperactivity, decay and distress. However, there is no significant correlation between burden and patient behaviours of passivity and repeated vocalizations. This tells us that the presence of behavioural disorders in the person with dementia has increased the burden of the caregiver, excluding acts of passivity and repeated vocalizations (Table 7).

### 3.7. Burden and Stage of the Disease

The study data indicate that the level of burden in the caregiver would also be related to the stage of the disease in which the patient is located (F = 15.70; *p* = 0.000). As shown in Table 8, the highest level of burden appears in the moderate stage of the disease.

### 3.8. Burden, Behavioural Disorders and Stage of Disease 

Analyzed data may indicate that the behavioural disorders that could generate the greatest overload on the caregiver may vary throughout the different phases of the disease (Table 9). In the mild stage, aggression, decay and depression behaviors could correlate with greater burden on the caregiver while in an intermediate phase of the disease the behaviors that might correlate with greater overload are aggression and irritability. In the severe stage, irritability seems to be the behaviour that might correlates with greater burden on the caregiver.

### 3.9. Abuse and Behavioural Disorder

Analyzed data may indicate that abusive behaviors could be associated with the presence of conduct disorders suffered by people with dementia (F = 9.35; *p* = 0.000). The behavioral disorders that mostly generate abuse seems to bee aggression, euphoria, lack of inhibition, irritability and motor hyperactivity (Table 10). 

### 3.10. Abuse and the Stage of the Disease

Following, we analyzed whether the abusive behaviors could vary depending on the stage of the patient´s disease. The results may indicate that abusive behaviors could be more frequent in certain phases of the disease (F = 44.310; *p* = 0.000) (Table 11). Specifically, it seems that abusive behaviors may occur more in the moderate phase of the disease. 

### 3.11. Abuse, Behavioural Disorders and Stage of the Disease

Analyzed data might indicate that the behavior disorders that generate the greatest overload on the caregiver could vary throughout the different phases of the disease. In the first phase, aggression, decay and depression behaviors might correlate with greater burden on the caregiver while in an intermediate phase of the disease the behaviors that might correlate with greater overload could be aggression and irritability. In the final stage, irritability continues to be the behavior that might correlates with greater burden on the caregiver (Table 12).

## 4. Discussion

The main aim of this study was to analyse the burden suffered by the main caregiver of people with dementia. Another objective was to know if this burden would be mediated by others variables such as behavioural disorders and stage of the disease.

With regard to the kinship between caregiver and person with dementia, 51.50 per cent of the caregivers surveyed in this study reported that they carried out the care of their husband or wife and 39.40 per cent who cared for their father or mother. Kaizik et al. [42] found in their research that the offspring as primary caregiver accounted for 21 per cent of the cases. Mahoney et al. [43] revealed that 44.40 per cent of the total caregivers included in the study were the child, while the figure of the spouse as the primary caregiver, accounted for only 18.30 per cent of the cases. Therefore, through the results obtained in this research, a change in trend could be seen in the role of the primary caregiver, indicating an increase in care by the men.

In this study, of the total number of informal caregivers who participated, 54.50 per cent were women, and the remaining 45.50 per cent were men. This data contrasts with the results of most reviewed research, which place the informal caregiver profile traditionally linked to women as the most common, in a highly polarized manner [44]. In the research carried out by Pérez-Rojo et al. [45], the percentage of men caregivers was 20.50 per cent and the percentage of women caregivers was 79.50 per cent. According to previous studies, one of the first findings of this work could be that there is an increase in the presence of male caregivers. However, while the task of caring could be more balanced compared to other revised studies, it seems to be focus on the female sex. This could be a reflection of the changes experienced in recent decades, with the greater incorporation of women into social and work life and the greater participation of men in household work and care, focusing on equity among the sexes. 

For the development of this research, informal carers of people with dementia, have not been institutionalized whose care has depended exclusively on the work and involvement of their own families. However, this situation involves the progressive and continuous emergence of multiple stressors for these families, which could generate conflict among their members, situations of disagreement and enhance those already existing problems in the family nucleus. In addition, the situation of family cohesion may be further deteriorated in cases where care falls almost entirely on the figure of a single informal caregiver taking into account, above all, that 80 per cent the patients care depends on the family and re-quires an almost entirely dedication [46,47].

The data of our study reported that 27.30% of the main caregivers showed light burden and the 18.20% of the main caregiver reported burden. Moreover, 51.50% of those main caregivers have admitted abuse. Pérez-Rojo et al. [45], and Henderson, Buchanan and Fisher [48] identified burden as a determining and differential factor in the onset of abuse. 51.50 per cent of the caregivers included in their study gave positive results on the Caregiver Abuse Scale (CASE), obtaining values of abuse similar to those detected by Pérez-Rojo et al. [45], in its investigation, in which 60 per cent expressed abuse. These data also coincide with the results obtained by Wiglesworth et al. [49] in its study, in which 47.30 per cent of dementia patients included had endured abuse. Therefore, the presence of burden could be closely linked to abuse, thus coinciding with what was stated by Serra et al. [50] in their study. These results may reflect the existence of a real problem, such as abuse of people with dementia residing at home, whose detection may be going unnoticed and which, due to the increased vulnerability of these patients. In order to reduce the burden in caregivers, several studies focused on factors that may provide some protection against the experience of burden in them. Caregiver education [51], the provision of practical support, the presence of adaptive coping mechanisms [52] and an extraversion personality [53] have showed to prevent caregiver from burden.

As for caregiver burden, how the presence of different behavioural disorders influences their onset has been analysed in this study. The results may reflect a significant relationship between the onset of caregiver burden and behavioural disorders in the person with dementia, with the exception of passivity behaviour and repeated vocalizations. According to Van der Lee et al. [22], the presence of behavioural disturbances is the aspect that generates the greatest burden in caregivers, being one of the most stressful traits derived from the care of people with dementia. Burden creates a negative impact on the health of the caregiver but also on the care that they provide. In addition, on the one hand these authors detected situations of burden in 79 per cent of cases in this study that included the presence of behavioural disorders, thus making this relationship clear and coinciding with what we found in this study. On the other hand, Torrisi et al. [54] stated that the effect of behavioural disorders on caregiver burden was even greater than that generated by cognitive decline or functional limitations caused by dementia. Also, Hessler et al. [55] indicated that the aggressiveness and irritability behaviours were the most stressful for caregivers, coinciding with the results obtained in this investigation, in which the patient’s behaviours of agression, irritability and lack of inhibition may have a stronger association with the appearance of burden in the caregiver. 

Another aspect studied in this research focused on the variability that caregiver burden may manifest, depending on the stage of the dementia. Caregivers who participated in this study reported higher levels of burden in the moderate stages of the disease, with light burden being in 35 per cent of cases and burden in 25 per cent in this phase. In the same way, Black et al. [56] and Mohamed et al. [57] obtained similar results in their studies. Nevertheless, they linked this increase of burden in moderate stage dementia to the advancement of the cognitive decline and the consequent increased dependence on the caregiver. Furthermore, these informal caregivers have not received adequate training to carry out their task and adapt to the changes required by this demanding role [58]. However, these results would lead to believe that the increasing in cognitive decline and dependence in the late stages of the disease should result in a significant increase in caregiver burden during this phase. Nonetheless, far from this claim, the lowest levels of burden in this study were detected in situations where the patient was in the final stage of the disease. The answer to this contrast, as indicated by Givens et al. [59] in their research, could be linked to the presence of behavioural disorders, such as behaviour of lack of inhibition, aggression or irritability on the part of the patients, which contribute to stress and burden of the caregiver, underlying their health and well-being.

Taking these data into account, we noticed that behavioural disorders might play an important role in the care of people with dementia. For this reason, we focused the next step of our study on the analysis of differences in the frequency of occurrence of behavioural disorders in the different stages of dementia. The results obtained in this study seem to place the moderate stage of dementia as the phase of the disease in which more behavioural disorders could manifest, highlighting an increase in the behaviours of apathy, irritability and aggressiveness compared to the other phases. However, depression tends to appear more in early stages, as demonstrated by Onyike [60] in their study, which complement with the results obtained in our research. Although, some researchers suggested there was a link between depression and dementia, yet the nature of this relationship has not been determined. The symptoms of depression in early stages could be considered as response to the multiple psychological stress involved with dementia [61]. On the other hand, some authors consider that neuropsychiatric symptoms preceded the onset of cognitive difficulties in dementia. Our data, where depression symptoms appear in the mild stage of the disease, could be according with them. A potential mechanism for this may be found in the excessive glucocorticoid induction associated with depression, which in turn may result in a cascade of neurotransmitter-induced excitotoxicity, leading to subsequent neuronal injury dementia [62]. According with this study, depression could be viewed as a prodromal symptom for a dementing illness, related to early involvement of cerebral circuits crucial for mood regulation.

On the other hand, patients who were in the late stages of dementia seem to have few behavioural disorders and the caregivers might show less burden. Moreover, Kratz [63] emphasizes that behavioural alterations are related to the progression of cognitive decline linked to the evolution of dementia and what is more, Majer et al. [64] state that the fewer cognitive functions preserved the more frequent and severe of the occurrence of these behaviours. Therefore, according to these authors, the onset of behavioural disorders and burden could be more linked to the final stages of dementia. In contrast, the results obtained in our study could differ again from their conclusions reaching the highest levels in terms of the frequency of behavioural disorders being during moderate stages. However, these authors did not include a variable specifically related to disease phases in their studies, giving an overview of the progression process of the disease in its association with behavioural disorders, unlike the previous study a thorough analysis of behavioural alterations has been carried out specifically based on the different stages of dementia in this investigation.

Another important point of this study was analysed under which situations the main caregiver would manifest abuse. In this way, we tested the effect of behavioural disorders manifested by dementia sufferers and their relationship to abuse. As stated by Yan [65], the results obtained in this investigation might replicated this association, with the exception of the behaviour of passivity and repeated vocalizations. Regarding this, and related to Khoo et al. [66], the aggressiveness behaviour exhibited by people with dementia seemed to be the most closely related in this study to the increased abuse. The second seemed irritability behaviour, which displayed a strong link of abuse by the caregiver. Besides, some authors such as Cooney, Howard and Lawlor [28] pointed the presence of behavioural disorders as a risk factor for abuse in their research.

The results of this study related to abuse by the caregiver also might present differences in the development stages of dementia. The appearance of abuse by the caregiver could maintain a similar trend to that mentioned earlier in this research about burden and behavioural disorders, that is to say an increased presence of abuse identified in this study at the moderate stages of the disease. Positive results might be detected in the Caregiver Abuse Scale (CASE) for the risk of abuse in 70 per cent of caregivers of those patients who were in the intermediate stages of dementia. Owing to this, our results may reveal that behavioural disorders could play an important role for the onset of burden and abuse among caregivers of people with dementia, regardless of the severity of the cognitive impairment of the patients, thus coinciding with what was shown by Serra et al. [50]. That is, it is not so much the stress that occurs in the caregiver to meet the basic needs of the patients deal with them, but that the increased stress that is generated in caregivers could be linked to the presence of behavioural disorders. It should be remembered that, for the development of this research, informal carers of people with dementia, have not been institutionalized whose care has depended exclusively on the work and involvement of their own families. However, this situation involves the progressive and continuous emergence of multiple stressors for these families, which can generate conflict among their members, situations of disagreement and enhance those already existing problems in the family nucleus. In addition, the situation of family cohesion may be further deteriorated in cases where care falls almost entirely on the figure of a single informal caregiver taking into account, above all, that 80 per cent the patients care depends on the family and requires an almost entirely dedication [46,47].

There are certain limitations of this study that have to be taken into consideration. Given the correlational nature of our study, we cannot make any statements about causation. It would be useful to conduct a longitudinal study to confirm that caregiver burden and abuse could be mediated by behavioural disorders and the stage of the disease. Another limitation of our study is the size of the sample. Also, economic situations, cultural factors, personality factors and other variables that could affect caregivers were not controlled in this study.

However, our study has several strengths and advantages compared to previous studies. This research link burden in caregiver with another important variables like abuse, behavioural disorder and stage of the disease. Due to this study is intended to give visibility to the daily life work of these non-professional caregivers through the study of both their situation and the impact of certain aspects of dementia on their work. In this way, it is intended to warn of others truly complex situations, such as burden and abuse, which can go unnoticed by both healthcare professionals and society at large.

## 5. Conclusions

The data of our study revealed that non-professional caregivers of people with dementia could suffer from burden. Also, we found a correlation between burden and abuse. An increase in the level of burden seems to be followed by an increase in the level of abuse. Furthermore, we analysed several conditions that could have a correlation both with burden and abuse. We found that the presence of behavioural disorders, like aggression and irritability at the middle stage of the disease could be linked with burden in the caregiver. In addition, the presence of aggression, lack of inhibition and irritability could have a correlation with an increase in the probability that the people with dementia suffer abuse in the middle stage of the disease.

Considering that this study has some limitations mainly related to sample size and the absence of a follow-up of the patients and caregivers who have participated, it would be necessary to replicate it with a larger sample to compare the results. Moreover, it is necessary to raise awareness between society and health professionals of the effects that disabling diseases such as dementia have on their caregivers and family members, as well as the importance of caring for the carers and motivate the creation of future research projects in this area. 

## Figures and Tables

**Table 1 ijerph-18-01263-t001:** Demographic data of study sample (*n* = 33).

DEMOGRAPHIC DATA	% (*n*)
**Sex caregiver**	
**Men**	45.50 (15)
**Women**	54.50 (18)
**Sex patient**	
Men	30.30 (10)
Women	69.70 (23)
**Kinship with the patient**	
Couple	52 (17)
Siblings	3 (1)
Nephew/Niece	3 (1)
Offspring	39 (13)
Grandchild	3 (1)
**Type of dementia**	
Alzheimer D.	37 (12)
Mixed D.	9 (3)
Vascular D.	18 (6)
Senile D.	24 (8)
Parkinson D.	12 (4)
**Stage of dementia**	
Mild	21.20 (7)
Moderate	60.60 (20)
Severe	18.20 (6)

Alzheimer D.: Alzheimer Dementia; Mixed D.: Mixed Dementia; Vascular D.: Vascular Dementia; Senile D.: Senile Dementia; Parkinson D.: Parkinson Dementia.

**Table 2 ijerph-18-01263-t002:** Caregiver Burden (*n* = 33).

Burden Level	% (*n*)
Non-Burden	54.5 (18)
Light Burden	27.3 (9)
Burden	18.2 (6)

**Table 3 ijerph-18-01263-t003:** Caregiver Burden and Gender (*n* = 33).

Caregiver Burden	% (n)	
	Men (*n* = 15)	Women (*n* = 18)
Non-Burden	80 (12)	50 (9)
Light Burden	13.3 (2)	22.2 (4)
Burden	6.7 (1)	27.8 (5)

**Table 4 ijerph-18-01263-t004:** Caregiver Abuse Screen (*n* = 33).

Abuse	% (*n*)
Non-Abuse	48.50 (16)
Abuse	51.50 (17)

**Table 5 ijerph-18-01263-t005:** Behavioural Disorders.

Behavioural Disorders	Mean	SD
Aggression	3.121	3.638
Depression	4.242	4.000
Anxiety	3.515	3.930
Euphoria	2.121	3.039
Apathy	5.515	3.726
Lack of inhibition	2.090	3.136
Irritability	3.969	3.704
Motor hyperactivity	2.485	3.692
Decay	3.909	3.964
Anguish	4.424	3.992
Passivity	5.212	3.855
Repeated vocalizations	3.394	4.085

**Table 6 ijerph-18-01263-t006:** Relation between burden in caregiver and abuse (*n* = 33).

		Abuse % (*n*)	Non Abuse % (*n*)
Patient		51.50 (17)	48.5 (16)
Burden	Non-Burden	29.40 (5)	81.25 (13)
	Light Burden	35.30 (6)	18.75 (3)
	Burden	35.30 (6)	00.0 (0)

**Table 7 ijerph-18-01263-t007:** Correlations between Burden and Behavioural Disorder.

Behavioural Disorders	Pearson Correlation Coefficient (r)	*p*
Aggression	0.669	0.000
Depression	0.395	0.023
Anxiety	0.404	0.020
Euphoria	0.346	0.048
Apathy	0.219	0.022
Lack of inhibition	0.470	0.006
Irritability	0.624	0.000
Motor hyperactivity	0.377	0.030
Decay	0.363	0.038
Anguish	0.394	0.023
Passivity	0.147	0.413 (NS)
Repeated vocalizations	0.152	0.399 (NS)

NS: non-significant. ***p***: probability.

**Table 8 ijerph-18-01263-t008:** Burden and stage of the disease (*n* = 33).

	Stage of the Disease %(*n*)		
Urden (*n*)	Mild	Moderate	Severe
**Non-Burden**	71,4 (5)	40 (8)	83,3 (5)
**Light Burden**	14,3 (1)	35 (7)	16,7 (1)
Burden	14,3 (1)	B25 (5)	0 (0)

**Table 9 ijerph-18-01263-t009:** Correlations between Burden, Behavioural Disorders and stage of disease.

Stage of the Disease	Behavioural Disorders	r	*p*
Mild	Aggression	0.755	0.050
	Depression	0.840	0.018
	Decay	0.827	0.022
Moderate	Aggression	0.577	0.008
	Irritability	0.600	0.005
Severe	Irritability	0.851	0.032

r: Pearson correlation coefficient.

**Table 10 ijerph-18-01263-t010:** Correlations between Abuse and behavioural disorders.

Behavioural Disorders	r	*p*
Aggression	0.824	0.000
Depression	0.420	0.015
Anxiety	0.349	0.046
Euphoria	0.527	0.002
Apathy	0.223	0.212 (NS)
Lack of inhibition	0.657	0.000
Irritability	0.845	0.000
Motor hyperactivity	0.485	0.004
Decay	0.426	0.013
Anguish	0.384	0.027
Passivity	0.226	0.206 (NS)
Repeated vocalizations	0.230	0.198 (NS)

r: Pearson correlation coefficient NS: non-significant.

**Table 11 ijerph-18-01263-t011:** Abuse in the different stage of the dementia.

Stage of the Disease	Mean	SD
Mild	3	3364
Moderate	4.700	2.861
Severe	0.666	1.213

**Table 12 ijerph-18-01263-t012:** Correlations between Abuse, behavioural disorders and stage of the disease.

Stage of the Disease	Behavioural Disorders	r	*p*
Mild	Aggression	0.905	0.005
	Lack of inhibition	0.833	0.020
	Motor hyperactivity	0.851	0.015
Moderate	Aggression	0.732	0.000
	Lack of inhibition	0.571	0.009
	Irritability	0.827	0.000
Severe	Irritability	0.944	0.005

r: Pearson correlation coefficient.

## Data Availability

The data presented in this study are available upon request from the corresponding author.

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
