# Peer review of "Relation among Caregivers’ Burden, Abuse and Behavioural Disorder in People with Dementia"

_ijerph, 2021, doi:10.3390/ijerph18031263_

Round 1
Reviewer 1 Report
This study aimed to analyse whether the manifestation of behavioural disorders on people with dementia is related to the presence of burden in the informal primary caregiver using the Scales of Zarit, CASE and FAST. The results showed that the presence of behavioural disorders in patients with dementia were related to the increase in both the main caregiver burden and the risk of manifesting behaviours of mistreatment by the caregiver towards the patient, with both conditions appearing more frequently during the middle stages of the disease. There are some concerns as listed in the following:
(1) Correct the data in Tables 3, 4, 5, 6, i.e. 51,5 -> 51.5 and etc. in Table 3; 0,669-> 0.669 in Table 4; 71,4->71.4 in Table 6; 31,85->31.85 in Table 6.
(2) It is unclear why the data of Burden (%) in Table 6 were not consistent with the data (Burden + Intense Burden) in Table 5, e.g. Early: 31.85 vs. 28.6 (14.3+14.3); Middle: 43.90 vs. 60 (35.0+25.0); Late: 27.66 vs. 16.70 (16.70+0.00)
(3) How to get the conclusion for ”The results obtained from the Behaviour
Disorders Questionnaire indicate that behavioural disorders that generate the greatest burden in the caregiver appear, to a greater extent, in the middle phase of the disease.” (L288-290) from Figure 1. [clearly indicate aggregation and irritability, because the total mean? scales of the first 6 items are very close between early and middle groups; the last 2 items are excluded as they are NS in Table 4]
(4) typos and others:
*L18: the risk of manifesting behaviours of mistreatment towards the caregiver?
L305: Due to this this study
L326: Mahoney et al. [40], revealed
*L356-357: In addition, in our investigation, the 35.5 per cent of caregivers are found to be more likely to ill-treatment. -> from which result? It is 35.3+35.3 in Table 3.
*L368: with the exception of the behaviour of apathy, passivity and repeated vocalizations. -> apathy is significant as described in the section of 3.4. Burden and Behavioural Disorders.
L444: future research is remains needed to further analyse
*L469-: References: It is better to keep one format for the title (e.g. R6, 7, 30, 34, 39, 48, 49, 51, 53, 54), the journal (e.g. R3 vs. R7 and so on), the page number (e.g. R3: 637-651 vs. R4: 339-45)
Reviewer 2 Report
The work studies the relationship between caregivers’ burden, ill-treatment, and behavior disorder for specified patients. The author applied ZARIT interview, Functional Assessment Staging and Caregiver Abuse Screen to deliver data and results. 33 caregivers participated in the research, which is a merit of the work.
Overall the paper addresses an interesting problem; however, there are several major points that should be considered.
The authors discuss several papers in the introduction but without any comparisons. What is the contribution of the proposed work compared to the cited prior works? The contribution of using conventional methods in a similar but different topic is minor.
The conventional Zarit Burden Interview is controversial. You may need extra explanation since caregiver's burden is an inevitable fact. As caregivers, they are bound to face extraordinary burden. In addition, there is no explanation on criteria to distinguish terms ‘No burden’, ‘Burden’, and ‘Intense Burden’. Saying if your reader is not familiar with the Zarit Burden framework, they might be confused cause of your brief statements.
Moreover, it requires extensive explanation on the data stream, in the main body of the manuscript, instead of summarize everything in the discussion & conclusion part.
It is not clear how to identify the states of ‘ill-treatment’. The stated method is but subjective. I would suggest the authors put any intermediate results (if any) in the content or an appendix.
Why ‘No risk of ill-treatment’ in your table 3 brings about 81.3+18.8=100.1? Please correct. Why not use 81.25 and 18.75 instead?
Tedious discussion, but few remarkable highlights and new findings. It requires deeper investigation on the so called ‘Relation among caregivers´ burden, ill-treatment and behavioral disorder in people with dementia’. For example, the caregivers with social positive personality are more likely to agree to participant the survey, at the same time, they prone to take warm response to care the patient. How would you resolve the bias in research? How could you tell whether the participants’ feedbacks are reliable?
In summary, the problem studied is interesting and the idea & methods seems interesting although the finding and contribution seems to me minor at the moment.
Reviewer 3 Report
This is a full-length research report of identifying non-professional burden observed in informal caregivers who take care of people with various types of dementia and comparisons of the burden between characteristics of dementia in order to reveal the severity of burden observed in informal caregivers in terms of non-professional human burden in line with daily life. The authors have tried to reveal the factors as indexed by scores via some reliable measures of questionnaire towards informal caregivers. Based on data from the questionnaire, the authors revealed a significant association of the burden with the severity of the disease, especially behavioral disorders appeared in patients with dementia. As the authors described in the manuscript despite the limitations of the study, the purpose of the present study as well as the results is believed to contribute the re-consideration of professional healthcare system(s) and their daily burden. This issue has progressively been focused worldwide, and overall impact of their research seems to be considered strong.
I have following comments/suggestions to consider the revision:
- To enhance the strength of the manuscript, the authors should add to the manuscript their perspective for differences of degree of burden among the severity of dementia. Most importantly the readers wish to know the perspective as a take-home message of the present study.
- Is there any difference between the sex of the patients with dementia in terms of burden observed in informal caregivers? Differences of burden in terms of sex seem to be a predominant factor to be considered. Vice versa any difference between the sex of the caregivers?
- Please identify the exact name of Ethics Committee that had approved this study and also the approved number(s) assigned to this study.
Round 2
Reviewer 2 Report
The authors have responsed to the concerns, and the manuscript has been revised.
To my best understanding, the authors applies conventional approach to handle an interesting problem. They did some investigation., and conducted certain analysis. However, it seems that the investigations are very subjective and the analysis remains on the surface of the topic, thus limited scientifical contribution can be credited.
I would suggest the authors at least adds more in-depth analysis, for all three methods-subsections. Then, I would agree to recommend this work to be accepted by the journal.
Reviewer 3 Report
The authors have done a good job responding to reviewer comments and concerns in their revision. I believe the manuscript is significantly improved as a result.